# Antimicrobial agents for the treatment of enteric fever chronic carriage: A systematic review

Naina McCann[1,2]*, Peter Scott[1,2], Christopher M. Parry[3,4,5], Michael Brown[2,6]

1 UCL Faculty of Population Health Sciences, University College London (UCL), London, United Kingdom,
2 Hospital for Tropical Diseases, University College London Hospitals NHS Foundation Trust, London, United Kingdom, 3 Clinical Sciences, Liverpool School of Tropical Medicine, Liverpool, United Kingdom, 4 Alder Hey Children's NHS Foundation Trust, Liverpool, United Kingdom, 5 Centre for Tropical Medicine and Global Health, University of Oxford, Oxford, United Kingdom, 6 Clinical Research Dept, London School of Hygiene & Tropical Medicine, London, United Kingdom

* Naina.mccann@nhs.net, zchaf30@ucl.ac.uk

**Data Availability Statement:** All relevant data are within the paper and Supporting information files.

**Funding:** The author(s) received no specific funding for this work.

## Abstract

### Background

Chronic carriage of *S.* Typhi or *S.* Paratyphi is an important source of enteric fever transmission. Existing guidance and treatment options for this condition are limited. This systematic review aims to assess the evidence concerning the efficacy of different antimicrobials in treating enteric fever chronic carriage.

### Methods

We searched major bibliographic databases using relevant keywords between 1946 and September 2021. We included all interventional studies that included patients with confirmed enteric fever chronic carriage and deployed an antimicrobial that remains in clinical practice today. Case reports and case series of under 10 patients were excluded. Two reviewers screened abstracts, selected articles for final inclusion and quality-assessed the included studies for risk of bias. Extracted data was analysed, with pooling of data and eradication rates for each antimicrobial calculated. As only one randomised controlled trial was identified, no meta-analysis was performed.

### Results

Of the 593 papers identified by the initial search, a total of eight studies met the inclusion criteria and were included in the systematic review. Evidence was identified for the use of fluoroquinolones and amoxicillin/ampicillin in the treatment for enteric fever chronic carriage. Fluoroquinolones were superior to amoxicillin/ampicillin with 92% of patients achieving eradication after one antimicrobial course compared to 68% (p = 0.02). The quality of included studies was poor, and all were carried out before 1990.

**Competing interests:** The authors have declared that no competing interests exist.

## Conclusion

This review identified fluoroquinolones and amoxicillin/ampicillin as treatment options for enteric fever chronic carriage, with fluoroquinolones the more effective option. However, this evidence pre-dates rises in antimicrobial resistance in enteric fever and therefore the significance of these findings to today's practice is unclear. Further research is needed to investigate whether these antimicrobials remain appropriate treatment options or whether alternative interventions are more effective.

## Introduction

Enteric fever is a systemic febrile illness caused by infection with the Gram-negative bacteria *Salmonella enterica* subspecies serovars Typhi (*S.* Typhi) and Paratyphi A, B or C (*S.* Paratyphi). It causes significant morbidity and mortality globally, with approximately 14 million cases and 135,000 deaths per year, with the highest number of reported cases from South and South-East Asia [1].

Enteric fever is transmitted via the faecal-oral route via the ingestion of food or water contaminated with infected human faeces. Clinical illness usually occurs 7–14 days after exposure with a non-specific febrile illness. The majority of patients recover from acute enteric fever following an appropriate course of antimicrobials. Around 10% of patients continue to excrete *S.* Typhi or *S.* Paratyphi in their stool for a few weeks in the convalescent period following recovery from acute infection with approximately 1–5% of patients continuing to excrete *S.* Typhi or *S.* Paratyphi in their stool for more than one year [2, 3]. This latter group are known as chronic carriers [4].

The pathogenesis of chronic carriage is not well understood. The gallbladder and biliary system appear to be the site of primary persistence of *S.* Typhi and *S.* Paratyphi in chronic carriers and indeed those with gallstones are more likely to become carriers [2, 4, 5].

*S.* Typhi and *S.* Paratyphi are human-restricted pathogens and therefore chronic carriage plays an important role in maintaining the reservoir of infection in humans. Asymptomatic chronic carriers unknowingly transmit the disease to others by faecal contamination of food and water. The first described, and perhaps most famous example of this was Mary Mallon ('typhoid Mary') who worked as a chef in New York in the 1950s and infected at least 54 people as an asymptomatic carrier [3]. Since then forensic epidemiology has demonstrated many more such cases, for example Mr N the Folkestone milker who infected over 200 people over a number of years via infected milk [6].

In addition to the public health risk of chronic carriage there is evidence that chronic carriage is associated with an increased individual risk of malignancy, particularly gallbladder cancer [7–9]. A recent meta-analysis reported an overall odds ratio of gallbladder cancer in *S.* Typhi carriers of 4.28 (95% CI: 1.84–9.96) [10].

The identification and treatment of chronic carriers therefore has both a significant public health and arguably individual benefit. There is currently limited evidence or guidance on how these chronic carriers should be treated. Furthermore, recent clinical reviews on the subject give differing advice on whether antimicrobials are an effective treatment option for chronic carriage treatment [11–13].

WHO enteric fever guidelines from 2003 suggest treatment options for enteric fever chronic carriage of amoxicillin, co-trimoxazole or ciprofloxacin [14]. Over the last 30 years there have been significant changes in antimicrobial resistance patterns of *S.* Typhi and *S.*

Paratyphi, with decreased susceptibility to fluoroquinolones (Fq) now almost universal in some areas of South Asia and increasing across sub-Saharan Africa [15–17]. Multi-drug resistance (MDR), resistance to amoxicillin, co-trimoxazole, chloramphenicol, is also seen in around 40% of patients worldwide [15]. A current outbreak of extensively-drug-resistant (XDR) enteric fever in Pakistan has highlighted the limited antimicrobial treatment options available for acute enteric fever in this setting. The options for antimicrobial treatment of enteric fever chronic carriers in the era of drug-resistance are unknown.

The aim of this systematic review is to review the existing evidence of efficacy of antimicrobials in treating enteric fever chronic carriage. By doing so we hope to review existing knowledge and highlight areas for ongoing research and review.

## Methods

The reporting of this systematic review was guided by the standards of the Preferred Reporting Items for Systematic reviews and Meta-Analyses (PRISMA) statement [18]. A search of PROSPERO database performed did not reveal any existing similar protocols. The protocol for this review was not registered but can be found in supplementary material (S1 Protocol).

### Search strategy and selection criteria

A systematic search through MEDLINE, EMBASE and Web of Science from 1946 was initially performed on the 1st February 2021. The terms searched were ("typhoid" OR "paratyphoid" OR "salmonella typhi" OR "salmonella paratyphi "OR "enteric fever") AND ("chronic carriage" OR "disease carrier "OR "carrier state" OR "typhoid carrier" OR "paratyphoid carrier" AND ("antibiotic" OR "antibacterial" OR "antibacterial treatment" OR "antibiotic treatment" OR "antibacterial agent" OR "antibiotic agent" OR names of individual antibiotics). The full list of search terms is listed in the supplementary information (S1 Checklist). No language restrictions were included in the initial search. Reference lists and bibliographies of selected articles were also searched. A search for unpublished literature was not performed.

A repeat search of the same terms was performed in September 2021 to check for any additional studies that could be included prior to publication. All new articles identified that been published since the prior search date were reviewed by two independent reviewers.

We included studies that met the following criteria:

1. Baseline population of adults > 18 years with confirmed enteric fever chronic carriage

2. Intervention of an antimicrobial course

3. Outcomes of stool clearance measured as defined by authors of each included study

We did not include studies that assessed an antimicrobial that is no longer in clinical use today, for example sulphathiazole. Case reports and case series of under 10 cases were excluded to minimize study bias. If studies included participants with non-typhoidal salmonella (NTS) chronic carriage in addition to patients with enteric fever chronic carriage these studies were individually reviewed to assess whether they included 10 or more patients with enteric fever chronic carriage and, if they met other inclusion criteria, they were included. Studies that combined an antimicrobial treatment intervention with another intervention (e.g. cholecystectomy) were only included if they had an intervention arm that included antimicrobials alone and this was made up of at least 10 patients.

If journal articles were not available online, we requested print versions through store requests in our academic institution. Articles were excluded if we could not access a full-text

version for review. Two independent researchers screened titles and abstracts and reviewed full text articles for inclusion with any disagreements resolved by consensus.

## Quality of studies

Two independent reviewers evaluated the study quality with any disagreements resolved by consensus. Quality assessment was performed using the National Heart, Lung and Blood Institute quality assessment tools [19]. Each study was given an overall rating of good, fair or poor.

## Data extraction

Data extraction was performed using a standardized data extraction form. Data was collected on publication year, study country, study design, participant characteristics including number of patients with gallstones, number of participants given intervention, intervention type including dose and duration, stool culture result at follow-up and side effects of intervention.

## Data analysis

For each included study the proportion of those chronic carriers that eradicated the pathogen (eradication proportion) was calculated (i.e., the number of participants culture negative for *S*. Typhi or *S*. Paratyphi after an antimicrobial intervention divided by the total number of participants in the study).

For those that had a control population we calculated the eradication proportion in this population. Data were synthesised for studies using the same antimicrobial intervention and overall eradication proportions presented. The main outcome measure was eradication proportion after one course of antimicrobials. As some studies re-treated their patients' multiple times, we also calculated eradication proportion per antimicrobial course. Other data synthesised for each antimicrobial intervention included intervention characterises and side effects of antimicrobial use. Results were presented in tabular format comparing antimicrobial intervention. Categorical variables were compared using Fishers exact test. Data was collated in Microsoft Excel and analysed in RStudio version 1.4.1103.

# Results

## Flow of included studies

The initial search strategy identified 579 papers. The abstracts and titles were reviewed and after applying inclusion and exclusion criteria 26 papers were identified for full-text evaluation. Of these a further 18 were excluded for reasons shown in the PRISMA flow-chart in Fig 1, leaving 8 articles which met our inclusion criteria. Full-text manuscripts could not be accessed in 7 cases; all but one of these were published in non-English language journals (3 German, 1 Polish, 1 Russian, 1 Japanese). Of these 3 were published before 1960 and the remaining 4 were published before 1980. A further search using the same terms was performed in September 2021 which identified a further 14 papers, none of which were deemed eligible for inclusion.

## Study characteristics

Study characteristics of all included studies are summarised in Table 1. Only one used a blinded randomised-control trial (RCT) design. The remaining seven studies used an open, pre- post design method with no controls, comparing stool culture (outcome measure) in participants before and after treatment.

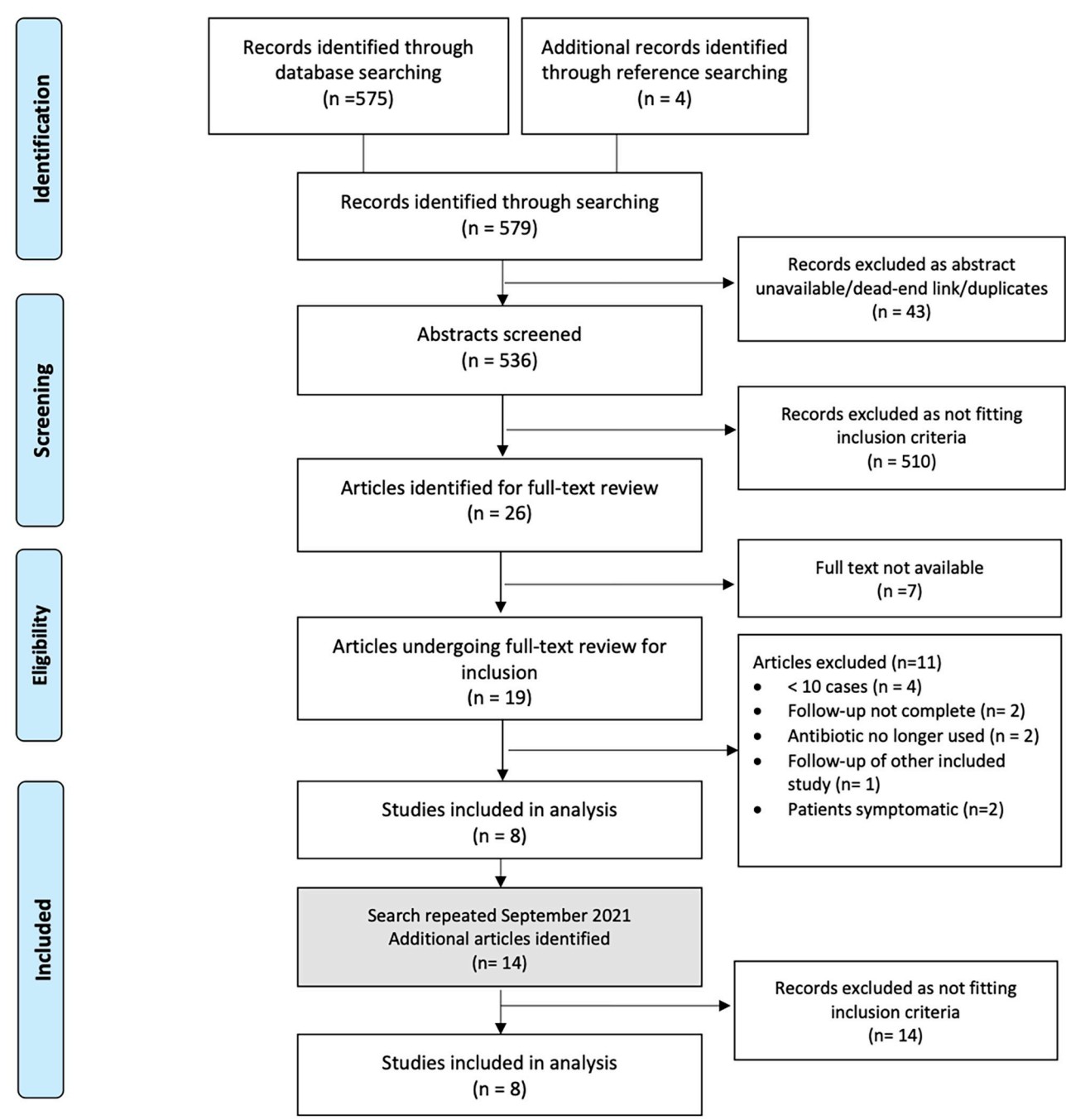

**Fig 1. Preferred Reporting Items for Systematic Reviews (PRISMA) flow diagram of the study selection process.** *From*: Moher D, Liberati A, Tetzlaff J, Altman DG, The PRISMA Group (2009). *Preferred Reporting Items for Systematic Reviews and Meta-Analyses: The PRISMA Statement.* PLoS Med 6(7): e1000097. doi:10.1371/journal.pmed1000097 For more information, visit www.prisma-statement.org.

Included studies were carried out between 1966 and 1988, with no studies included from the last 30 years. The majority of studies were carried out in the USA (n = 4) with the remaining papers conducted in Chile (n = 1), Peru (n = 1), Italy (n = 1) and Israel (n = 1).

Inclusion criteria for almost all the studies was presence of *S.* Typhi or *S.* Paratyphi in the stool or bile for greater than 12 months. One study necessitated prior history of acute enteric

**Table 1. Summary of included studies.**

| Study author and references | Country | Year | Journal published | Planned intervention, drug, dose and duration | Study design | Outcome measure | Quality assessment |
|---|---|---|---|---|---|---|---|
| Ferreccio et al. [20] | Chile | 1988 | Journal of Infectious Diseases | Ciprofloxacin PO 750mg BD, 28 days | Open pre- and post- trial, no control | 3 stool specimens cultured at 3,6,9 and 12 months post treatment | Fair |
| Gotuzzo et al [21] | Peru | 1988 | Journal of Infectious Diseases | Norfloxacin PO 400mg BD, 28 days | Double-blind randomised-controlled trial, followed by open trial | Stool culture at months 1,2,3,6,9,12 post treatment | Good |
| Phillips et al. [22] | USA | 1971 | Journal of American Medical Association | Ampicillin PO 1g QDS, 90 days | Open pre- and post- trial, no control | Stool culture monthly to a minimum of 6 months post treatment | Fair |
| Simon et al. [23] | USA | 1966 | New England Journal of Medicine | Ampicillin PO 75-100mg/kg/day for 28 days | Open pre- and post- trial, no control | Stool cultures monthly for first 3 months, 3 monthly for following year, then 2–3 times/year to a minimum of 7 months | Poor |
| Nolan et al. [24] | USA | 1978 | Journal of American Medical Association | Amoxicillin PO 2g TDS for 28 days | Open pre- and post- trial, no control | Stool culture at 3,6,12 months post treatment | Fair |
| Dinbar et al. [25] | Israel | 1969 | American Journal of Medicine | Ampicillin PO 5.25mg/day for 10–40 days | Open, multi-arm pre- and post- trial, no control | Stool cultures monthly for minimum of 12 months post treatment | Poor |
| Kaye et al. [26] | USA | 1967 | Annals of the New York Academy of Science | Ampicillin PO 1.5mg QDS + probenecid 0.5g QDS, 42 days | Open pre- and post- trial, no control | Stool cultures (regularity not defined) to a minimum of 12 months post treatment | Poor |
| Scioli et al. [27] | Italy | 1970 | Journal of Infectious Diseases | Ampicillin IV 1g TDS, 15 days | Open pre- and post- trial, no control | Stool cultures twice/week for 9 weeks, then once/month for a minimum of 16 months | Poor |

fever [21] and one study included two patients who had positive stool cultures with an elevated Vi antibody level, without need for repeated positive cultures of > 1 year [20]. One study mentioned that included patients had been previously treated for chronic carriage [27] but others did not specify this. Length of chronic carriage varied between participants within studies, with the longest carrier state being 39 years. Two studies excluded those with serious underlying health conditions and patients with penicillin allergies [21, 24]. Exclusion criteria were not well defined in the other studies.

The age range of included participants was broad, including adults from 18–81 years. One study included one child aged 7 years old with the results not separated from the 14 adults in the study [23]. After discussion between reviewers this study was included. The majority of participants in the studies were female which correlates with prior knowledge that females are more likely to be carriers [28].

Most of the studies investigated participants for underlying gallstones before enrolment in the study using ultrasound or cholangiography or X-ray. However, in some patients the biliary tract was not well visualised and two studies did not investigate for underlying gallstones prior to enrolment [20, 22]. These patients were categorised in a third category 'gallstone status unknown'.

## Study intervention

The intervention drug was a fluoroquinolone (Fq) in two studies: norfloxacin was used in one and ciprofloxacin in the other. Four of the studies used ampicillin alone as the intervention, one study used amoxicillin and one used ampicillin with probenecid. One of the ampicillin studies used intravenous administration but the rest of the studies used oral administration. Dosing varied between the different studies (see Table 1). One study also included separate

arms of the study with patients undergoing cholecystectomy and cholecystectomy in combination with ampicillin treatment [25]. In this study six of the patients undergoing antimicrobial treatment alone had previously undergone cholecystectomy in the other arm of the study. These were included as they were confirmed to still be chronic carriers 1 year post cholecystectomy and prior to enrolment in the antimicrobial arm.

## Quality assessment

Overall methodological quality of included studies was poor (see Table 1). There was only one randomised and blinded study, with the majority of studies designed as pre- vs post-studies without a control.

In the RCT study, the outcomes were inconsistently reported with the summary table showing no cases of eradication in the placebo group but the text reporting that one patient this arm did have negative stool cultures at follow-up (indicating spontaneous cure). Furthermore, 13 patients were enrolled in the norfloxacin arm but only 12 patients included in the final analysis with no intention-to-treat analysis performed. No relative risk or odds ratio was calculated.

The remainder of the studies were open, pre- vs post- studies without controls. Overall inclusion criteria were well defined, but studies did not clearly state how patients were enrolled.

Interventions were generally well defined but often not consistent across participants, with some studies changing dosing regimens during the study period and one study using multiple interventions on participants [25]. All studies clearly defined the outcome measure, and this was measured before and after intervention and at repeated intervals for adequate follow-up periods after intervention. Loss to follow-up in the included studies was small (less than 20% in all). None of the studies mentioned a power calculation and numbers of participants were small in all studies. Minimal statistical analysis was performed.

A meta-analysis was not performed as there was only one RCT identified.

## Outcomes by antimicrobial group

**Fluoroquinolones.** Two studies assessed Fq as a treatment option for *S.* Typhi chronic carriage, and no studies assessed Fq as a treatment option for *S.* Paratyphi chronic carriage. In the *S.* Typhi studies a total of 25 patients received an intervention course [20, 21]. The norfloxacin study was an RCT, comparing 28-days of 400mg BD norfloxacin to placebo. The ciprofloxacin study was a pre-post study investigating the effect of a 28-day course of 750mg BD ciprofloxacin. All patients had *S.* Typhi chronic carriage, 44% of whom were known to have gallstones (although this was not specifically looked for in one of the studies). Both Fq studies showed an eradication proportion of 92% following a single 28-day treatment course. These two studies are summarised in Table 2.

*Effect of gallstones on eradication.* Patients in the ciprofloxacin study were not routinely screened for biliary disease before enrolment in the study, however two patients were known to have gallstones. In the norfloxacin study patients underwent oral cholecystogram, IV cholangiography or gallbladder ultrasound before enrolment.

Eradication proportions were 100% in those without gallstones, 89% in those with unknown gallstone status and 82% in those with known gallstones.

*Side effects.* Side effects were seen in a quarter of patients. In the ciprofloxacin study two patients stopped treatment early due to side effects (haemoglobin drop and urticarial rash). Of note, three other patients in this study had a haemoglobin drop that were attributed to

**Table 2. Summary of results from studies using fluoroquinolone antibiotics to treat *S.* Typhi chronic carriage.**

| | *Gotuzzo* | | *Ferreccio* | *Combined interventions* |
|---|---|---|---|---|
| *Intervention given* | **Norfloxacin** | **Placebo** | **Ciprofloxacin** | |
| *Participant characteristics* | | | | |
| *Total no of participants, n* | 13 | 12 | 12 | 25 |
| *With gallstones, n (%)* | 9 (69) | 9 (75) | 2 (17) | 11 (44) |
| *Without gallstones, n (%)* | 4 (31) | 3 (25) | 0 | 4 (16) |
| *Unknown gallstone status, n (%)* | 0 (0) | 0 | 10 (75) | 10 (36) |
| *Demographics* | | | | |
| *Age, mean* | 36 | 35 | 31 | 34 |
| *Age, range* | 18–58 | 12–67 | 20–51 | 18–67 |
| *Female, n (%)* | 6 (46) | 11 (92) | 10 (83) | 16 (64) |
| *Microbiological characteristics* | | | | |
| *S. Typhi, n (%)* | 13 (100) | 12 (100) | 12 (100) | 25 (100) |
| *MIC to intervention drug, range (ug/mL)* | 0.06–0.5 | 0.06–0.5 | 0.0156–0.0078 | 0.0078–0.5 |
| *Intervention* | | | | |
| *Administration method* | PO | PO | PO | |
| *Total daily dose, mean (mg)* | 800 | NA | 1500 | |
| *Total duration, mean (days)* | 28 | 28 | 25 | |
| *Duration range, (days)* | 28 | 28 | 10–28 | |
| *Outcomes—eradication rates* | | | | |
| *Minimum follow-up (months)* | 3 | 3 | 12 | 3 |
| *Total no of patients included in analysis[1], n* | 12 | 12 | 12 | 24 |
| *Total no patients eradicated at end of follow-up, n (%)* | **11 (92)** | **1 (8)** | **11 (92)** | **22 (92)** |
| *Eradication rate in those with gallstones, n (%)* | 7 (88) | 1 (8) | 2 (100) | 9 (82) |
| *Eradication rate in those without gallstones, n (%)* | 4 (100) | 0 (0) | 0 (0) | 4 (100) |
| *Eradication rate in those with unknown gallstone status, n (%)* | 0 (0) | 0 (0) | 9 (90) | 9 (90) |
| *Outcomes—side effects* | | | | |
| *Total side effects, n (%)* | 1 (8) | 3 (25) | 5 (41) | 6 (25) |
| *Diarrhoea, n (%)* | 0 | 0 | 0 | 0 |
| *Rash, n (%)* | 0 | 0 | 1 (8) | 1 (4) |
| *Other, n (%)* | 1 (8) | 3 (25) | 4 (33) | 5 (21) |
| *Total no of antibiotic courses given[2], n* | 23 | 12 | 12 | 35 |
| *Cure rate per no of courses given, n (%)* | 18 (78) | 1 (8) | 11 (90) | 29 (83) |

[1]—One patient was excluded from norfloxacin analysis in Gottuzo paper due poor adherence

[2]—In Gotuzzo paper 10 placebo patients were re-treated with norfloxacin openly taking the total number of treatment courses given in this paper to 23

In the RCT the norfloxacin group had a higher proportion of *S.* Typhi eradication than the control group (92% vs 8%, p <0.001).

ciprofloxacin therapy. Norfloxacin was relatively well-tolerated in the other study with only one patient reporting a rash, although these are not further explored in this paper.

**Effect of repeated courses.** In the norfloxacin study patients initially treated with placebo were then retreated with norfloxacin. This group had a slightly lower eradication proportion of 70%, lowering the overall eradication proportion to 83% per antibiotic course overall.

**Ampicillin/amoxicillin.** The remaining six studies identified assessed amoxicillin or ampicillin (amox/amp) use. The results of these are summarized in Table 3.

Overall, 101 patients were given a total of 115 courses of amox/amp (either PO or IV). Most of these patients were female and 28% had confirmed gallstones. Only one patient had *S.* Paratyphi. The dosing regimens were relatively high with the mean daily dose of amoxicillin 4.5g/

**Table 3. Summary of results from studies using amoxicillin or ampicillin to treat enteric fever chronic carriage.**

| | Phillips | Simon | Nolan | Dinbar | Kaye | Scioli | Overall |
|---|---|---|---|---|---|---|---|
| *Participant characteristics* | | | | | | | |
| Total no of participants | 12 | 15 | 15 | 16 | 24 | 19 | **101** |
| With gallstones, n (%) | 0 | 5 (33) | 4 (27) | 3 (12) | 12 (50) | 5 (26) | **29 (29)** |
| Without gallstones, n (%) | 0 | 10 (66) | 11 (73) | 7 (87) | 7 (25) | 8 (42) | **43 (42)** |
| Unknown gallstone status, n (%) | 12 (100) | 0 (0) | 0 (0) | 6 (38) | 5 (21) | 6 (32) | **29 (29)** |
| Known to have had cholecystectomy, n (%) | 0 (0) | 1 (7) | 0 | 7 (44) | 1 (4) | 1 (5) | **10 (10)** |
| *Demographics* | | | | | | | |
| Age, mean | 57 | 44 | 65 | 53 | 59 | 45 | **54** |
| Age, range | 23–81 | 7–62 | 48–77 | 36–67 | 33–83 | 21–64 | **7–81** |
| Female, n (%) | 7 (58) | 10 (66) | 13 (87) | 7 (44) | 18 (75) | 16 (84) | **71 (70)** |
| *Microbiological characteristics* | | | | | | | |
| S. Typhi, n (%) | 12 (100) | 15 (100) | 12 (100) | 15 (94) | 24 (100) | 19 (100) | **100 (99)** |
| MIC to intervention drug, range (ug/mL) | NA | <1–2 | < 1 | 0.5–2.5 | NA | NA | |
| *Intervention characteristics* | | | | | | | |
| Intervention name | Ampicillin | Ampicillin | Amoxicillin | Ampicillin | Ampicillin | Ampicillin | |
| Administration method | PO | PO | PO | PO | PO | IV | |
| Intended daily regimen | 1g QDS | 75-100mg/kg | 2g TDS | 1.25g QDS | 1.5g QDS | 1g TDS | |
| Intended duration (day) | 90 | 28 | 28 | 10–40 | 42 | 15 | |
| Addition of probenecid | No | No | No | No | Yes | No | |
| *Dosing details of initial course given* | | | | | | | |
| Total daily dose, mean (g) | 4 | 4 | 5.2 | 5.1 | 5.5 | 3 | **4.5** |
| Total course duration, mean (days) | 90 | 28 | 28 | 25 | 36 | 15 | **37** |
| Total dose taken during course, mean (g) | 360 | 112 | 146 | 128 | 198 | 45 | **165** |
| *Outcomes—eradication* | | | | | | | |
| Minimum follow-up (months) | 18 | 7 | 12 | 12 | 12 | 16 | |
| **Total no patients eradicated after initial course, n (%)** | **9 (75)** | **13 (87)** | **11 (73)** | **7 (44)** | **9 (38)** | **19 (100)** | 68 (67) |
| Eradicated in those with gallstones, n (%) | 0 (0) | 4 (80) | 3 (75) | 0 (0) | 4 (33) | 5 (100) | **16 (55)** |
| Eradicated in those without gallstones, n (%) | 0 (0) | 8 (89) | 9 (82) | 4 (57) | 5 (71) | 8 (100) | **34 (79)** |
| Eradicated in those with unknown gallstone status, n (%) | 9 (75) | 0 (0) | 0 (0) | 3 (50) | 0 (0) | 6 (100) | **18 (62)** |
| Eradicated in those with known cholecystectomy, n (%) | 0 (0) | 1 (100) | 0 (0) | 3 (43) | 0 (0) | 1 (100) | **5 (50)** |
| Total no of patients eradicated at end of follow-up, n (%) | 9 (75) | 13 (87) | 11 (92) | 9 (56) | 12 (50) | 19 (100) | **73 (73)** |
| **Total no of antibiotic courses given[2], n** | 12 | 17 | 15 | 21 | 31 | 19 | **115** |
| No of antibiotic courses resulting in successful eradication (%) | 9 (75) | 13 (76) | 11 (73) | 9 (43) | 12 (39) | 19 (100) | **73 (63)** |
| *Outcomes—side effects* | | | | | | | |
| Total side effects, n (%) | 8 (67) | 10 (67) | 7 (47) | NA | 5 (21) | 3 (15) | **33 (33)** |
| Diarrhoea, n (%) | 5 (42) | 8 (53) | 3 (20) | | 1 (4) | 2 (10) | **19 (19)** |
| Rash, n (%) | 5 (42) | 6 (40) | 2 (13) | | 5 (21) | 1 (5) | **19 (19)** |
| Anaphylaxis | 1 (9) | | | | | | **1 (1)** |
| Other, n (%) | 0 (0) | 2 (20) | 3 (20) | | | | **5 (5)** |

day but this varied significantly between studies (range 3–5.5g/day). The mean duration of treatment was 37 days but again this was highly variable between regimens (range 15–90 days).

The overall eradication proportion after a single course of amox/amp was 68% but again, this was highly variable between studies (range 38–100%).

**Effect of administration method on eradication.** There was only one study looking at the effect of IV ampicillin on chronic carriage eradication proportion [27]. The mean daily dose and total cumulative dose of IV ampicillin used in this study were much lower than those

used in the oral studies (3g vs a mean of 4.7g for the oral studies). However, the eradication proportion in this study was notably higher when compared to the oral ampicillin studies (100% vs 60%, p <0.01) with no failures described in the IV arm.

**Effect of dosing regimen and duration on eradication.**   The total daily dose of amox/ amp varied between 4 and 5.5g for the studies using oral intervention, with the total drug given over the duration of the course between 112g to 360g.

Overall comparisons between dosing regimen and duration are not possible as different studies used widely variable dosing regimens and durations, often not standardised within their own study population. Therefore, individual regimens within studies were examined more closely to investigate for possible associations of dosing or duration with outcome.

In the Nolan et al study [24] two different dosing regimens of amoxicillin were given. For those that tolerated it (n = 10) a dosing regimen of 2g of oral ampicillin was administered three times daily (total daily dose of 6g) which had a 90% success at 12 months after one course of 28 days. In those who did not tolerate the higher dose (n = 5) a lower dosing regimen of 1g oral ampicillin three times a day (total daily dose of 3 grams) was started. This regimen only had a 40% eradication proportion at 12 months after 28 days, but the numbers were very small.

Similarly, in Phillips et al. [22] those who completed the full dosing schedule of 1g QDS for 90 days had a 90% eradication proportion whereas those who were unable to complete it had a 0% eradication proportion, although these numbers are small and the exact dosing regimens are not fully described in the paper.

In Dinbar et al. [25] the initial course length was 10 days with a total of 52g of ampicillin given over this time. This regimen had an eradication proportion of 38%. Over the period of the study the dose of ampicillin was increased to 200g given over 40 days. The eradication proportion of this regimen was 57%. There was no evidence to support a difference in eradication proportions between dosing regimens (p = 0.61).

The Kaye study, which have a mean daily dose of ampicillin of 5.5g for a mean duration of 36 days, was the only study to use probenecid in addition to ampicillin. Eradication proportions were lower in this study when compared to the other studies where probenecid was not used (38% vs 76%).

Overall, we are unable to draw any firm conclusions from this data whether dose or duration of this intervention effects eradication proportion although there is a slight suggestion that higher daily dosing and intravenous dosing may be slightly more effective.

**Effect of gallstones on eradication.**   All studies investigated for prior presence of gallstones except Phillips et al.

The eradication proportion was lower in those with gallstones compared to those without gallstones (55% vs 79%, p = 0.039). Those with unknown gallstone status had an eradication proportion of 62%, whereas those with a past cholecystectomy only had an eradication proportion of 50%, although the numbers in this group are small.

**Effect of repeated courses on eradication.**   After failing an initial course of therapy some patients were re-treated with a second or third course. In Kaye et al six patients who failed an initial course were re-treated with further courses of ampicillin. Of these only two were successfully eradicated [26]. In Dinbar et al four patients who initially failed therapy were successfully treated with higher dosing regimens of ampicillin [25]. In Simon et al one patient was re-treated twice with prolonged six-week courses of ampicillin which did not result in eradication [23]. In total, of the 11 patients that were re-treated, the eradication proportion was only 36%.

**Side effects.**   A third of patients in the ampicillin studies suffered side effects from their therapy. Rash and diarrhoea were the most common side effects and in two studies this

resulted in a modification to therapy [22, 24]. In the IV amoxicillin trial side effects were minimal. However, side effects were not routinely reported across the studies.

**Comparing antimicrobial groups.** Fluoroquinolones have a higher overall eradication proportion than amox/amp (92% vs 68%, p = 0.02). In those with gallstones the eradication proportion remains higher in those taking a Fq compared to those taking amox/amp but this is not significant (82% vs 55%, p = 0.16).

## Discussion

This is the first systematic review on the efficacy of antimicrobial agents for treatment of enteric fever chronic carriage. A total of 126 participants in 8 studies were included in this review and two intervention groups were identified: Fq and amox/amp. Overall eradication proportion was higher in the Fq compared to the amox/amp group (92% vs 68%, p = 0.02), but eradication proportions varied highly across the included studies.

These findings are consistent with the limited existing guidelines available on treatment of chronic carriage. As previously stated, the only available guidance for treatment of chronic carriers is from the WHO 2003 guidelines which suggests a quinolone, amoxicillin or co-trimoxazole [11, 14]. In this review we have not identified any specific evidence for use of co-trimoxazole. During the screening process we did identify literature investigating the use of co-trimoxazole in addition to other antimicrobials such as chloramphenicol, kanamycin and penicillin [29–33]. None of these studies met the criteria for inclusion in this study, again highlighting the poor quality of available evidence on this subject. Most of these studies were case reports, small case series or did not report on necessary outcome data.

The findings of this systematic review should be interpreted with caution. Firstly, the overall quality of the included studies was poor, with only one RCT included. The remainder of the studies were case series or pre- and post- studies with no control group and no blinding, which allowed for significant bias. Furthermore, in most studies it was unclear how patients had been identified for participation. Interventions were non-consistent within and across studies and the total number of included patients is small, limiting statistical analysis and overall conclusions.

Secondly, almost all the patients included in this study had *S*. Typhi chronic carriage and therefore limited conclusions can be made about the effectiveness of these antimicrobials on *S*. Paratyphi chronic carriage. With the rollout of the typhoid conjugate vaccine and the likely associated decrease in incidence of *S*. Typhi, the incidence of *S*. Paratyphi may start to rise [34]. Given there is currently no licensed vaccine for *S*. Paratyphi the role of treatment of carriers of this condition may become increasingly important in reducing enteric fever transmission and disease. This review identifies current large gaps in the evidence base for the treatment of this neglected condition.

Thirdly, and perhaps most importantly, the included trials were all carried out prior to widespread antimicrobial resistance of enteric fever. There are no studies that have been carried out to assess treatment options for enteric fever chronic carriage in patients with multi-drug resistant (MDR), Fq-resistant, or indeed extensively-drug-resistant enteric fever. This data is therefore inadequate to inform decisions regarding antimicrobial treatment of chronic carriage in the modern-day era of drug-resistant enteric fever.

Further research needs be done to understand and improve treatment options for enteric fever chronic carriers today. The first barrier to investigating treatment interventions on chronic carriers is the accurate identification of chronic carriers from the population. In the trials included in this study, regular stool cultures were used to identify carriers over many years. This is logistically challenging and not suitable for large-scale screening today,

particularly in low-resource endemic areas. Furthermore, as stool shedding is intermittent, the sensitivity is low [35]. Molecular and serological strategies to identify carriers have been explored but do not perform well, particularly in endemic settings [36, 37]. New methods to identify carriers such as antigen-specific antibodies and biomarkers show promise but need further exploration [38, 39].

Once carriers can be accurately identified, further work is needed to investigate the effect of different interventions on this population. There is limited data on the antimicrobial susceptibility of chronic carriage isolates and whether they harbour similar resistance profiles to acute enteric fever strains. A study from 2012 in Nepal looking at *S*. Typhi and *S*. Paratyphi isolates identified from cholecystectomies found lower rates of antimicrobial resistance in these strains, compared to acute enteric fever isolates in the same area [40]. This suggests that chronic carriage strains may originate from older, drug-susceptible strains for which Fq and Amp may still be effective. Previous work looking at the genetic diversity of S. Typhi has suggested this may be due to two different evolutionary methods within the population structure of S. Typhi [41].

Even if chronic carriage isolates do show reduced susceptibility to ciprofloxacin, the excellent bile penetration of this antimicrobial (reaching 2500–4500% of plasma concentrations in the bile [42]), may be sufficient to overcome the relatively low minimum inhibitory concentration (MIC) (e.g. with MIC $\leq 1$) isolates, although there is no patient outcome data to support this.

Alternative antimicrobials that have not been identified in this review may also be effective in treating chronic carriage. Azithromycin is currently used to treat uncomplicated enteric fever worldwide with good efficacy [43] but no studies have examined the efficacy of azithromycin on enteric fever chronic carriage. Azithromycin has good bile penetration [42] and a single case report suggests it may eradicate non-typhoidal *Salmonella* carriage [44]. It therefore may be a good intervention for enteric fever chronic carriage treatment and is currently the preferred treatment for Fq resistant chronic carriage eradication in the UK; post-treatment monitoring will be used to assess outcome.

It is well understood that the gallbladder is an important niche for the persistence of *S*. Typhi and *S*. Paratyphi and the development of carriage [3, 11]. Accordingly, cholecystectomy has been used as treatment for chronic carriage in the past [25, 45, 46]. In addition to carrying a significant anaesthetic and surgical risk this intervention does not guarantee elimination of the carrier state, with success rates reported between 70–90%. Studies in this review also identify multiple patients identified as carriers who had already undergone a prior cholecystectomy. Hence, it is likely that there are other foci of infection outside the gallbladder that also play an important role in *S*. Typhi and *S*. Paratyphi persistence and carriage, for example the biliary tree, liver or lymph nodes [47].

Recently, murine models have shown that biofilm formation, on gallbladder epithelium and cholesterol-rich gallstones, may play an important role in the development of carriage and protect against antimicrobial treatment [48–51]. Prior guidance similarly states that carriers with underlying gallstones are unlikely to be eradicated with antimicrobials alone [11, 14]. In contrast, this review shows relatively good eradication proportions in those with gallstones, particularly when Fq treatment was used, albeit lower than those without gallstones. Furthermore, many of the identified carriers in this review were not found to have gallstones, despite thorough investigation. It is evident that further work is required to better understand the pathophysiology of chronic carriage to allow the development of targeted interventions.

This review was limited by the high risk of bias in all included studies. Many studies showed selection bias with undefined selection of participants and exclusion of certain groups. The participants may not be a true representative of the carrier population. Minimal data were

reported from all studies with very little statistical analysis performed. Particularly, the presence of gallstones was not investigated or reported in all studies meaning conclusions on whether gallstones affect the effect of antimicrobial eradication are uncertain.

Side effects of medications were also poorly reported across studies. Some adverse events were linked to the intervention without obvious causality e.g. ciprofloxacin causing haemoglobin drop, when this is not a known common side effect. Furthermore, associations of Fq therapy that were not recognised when these studies took place, for example tendinopathy and aneurysm rupture, may have been under-reported. The recent realisation of potential significant side effects of Fq use has also led to restrictions in their use and they may no longer be suitable to treat patients with chronic carriage, again highlighting the need for alternative treatment strategies [52, 53].

In this review we excluded case reports or case series of less than 10 patients in attempt to reduce bias but in doing so may have excluded studies with additional interventions. Furthermore, the evidence relating to this topic is almost all more than 30 years old and some full-text articles were not able to be accessed, some of which may have otherwise been included in the final review. The strengths of our study include a comprehensive search strategy, the use of two independent reviewers throughout the process and the use of well-defined data extraction and quality assessment tools.

In conclusion, this review identifies evidence for fluoroquinolones and amoxicillin/ampicillin as antimicrobial interventions for treatment of enteric fever chronic carriage. Fluoroquinolones are the most effective antimicrobial intervention and should be recommended for first-line treatment of chronic carriage where the isolate is susceptible, given the lack of evidence for any other interventions. There is insufficient evidence to recommend empiric use of fluoroquinolones or amoxicillin/ampicillin for enteric fever chronic carriage today, due to significant changes in antimicrobial resistance over the last 30 years. Updated evidence on this neglected topic is required and should be a vital component of global eradication strategies for enteric fever going forward.

## Supporting information

**S1 Checklist. PRISMA 2020 checklist.**
(PDF)

**S1 File. Search strategy.**
(PDF)

**S1 Protocol. Systematic review protocol.**
(PDF)

**S2 File. Comparing categorical variables, R file.**
(PDF)

## Acknowledgments

The authors would like to thank Dr Daniel Davis for providing advice during the systematic review process and Dr Robert Shaw for proof-reading the manuscript.

## Author Contributions

**Conceptualization:** Naina McCann, Christopher M. Parry, Michael Brown.

**Data curation:** Naina McCann, Peter Scott.

**Formal analysis:** Naina McCann, Peter Scott.

**Investigation:** Naina McCann, Peter Scott.

**Methodology:** Naina McCann, Peter Scott.

**Supervision:** Christopher M. Parry, Michael Brown.

**Validation:** Christopher M. Parry, Michael Brown.

**Visualization:** Naina McCann.

**Writing – original draft:** Naina McCann.

**Writing – review & editing:** Naina McCann, Peter Scott, Christopher M. Parry, Michael Brown.

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
