## [Decision Letter · Decision Letter 0]

30 Mar 2022

PONE-D-21-36297Antimicrobial agents for the treatment of enteric fever chronic carriage: A systematic reviewPLOS ONE

Dear Dr.  McCann

Thank you for submitting your manuscript to PLOS ONE. After careful consideration, we feel that it has merit but does not fully meet PLOS ONE’s publication criteria as it currently stands. Therefore, we invite you to submit a revised version of the manuscript that addresses the points raised during the review process.

We look forward to receiving your revised manuscript.

Kind regards,

Praveen Rishi, Ph.D., FAMI, FABMS

Academic Editor

PLOS ONE

Journal Requirements:

Reviewers' comments:

Reviewer's Responses to Questions

**Comments to the Author**

1. Is the manuscript technically sound, and do the data support the conclusions?

Reviewer #1: Partly

Reviewer #2: Yes

2. Has the statistical analysis been performed appropriately and rigorously? 

Reviewer #1: I Don't Know

Reviewer #2: Yes

3. Have the authors made all data underlying the findings in their manuscript fully available?

Reviewer #1: Yes

Reviewer #2: Yes

4. Is the manuscript presented in an intelligible fashion and written in standard English?

Reviewer #1: Yes

Reviewer #2: Yes

5. Review Comments to the Author

Reviewer #1: The MS reports on  antimicrobial agents for the treatment of enteric fever chronic carriage: A systematicreview. In this review the author presented about the efficacyof different antimicrobials in treating enteric fever chronic carriage.   Author also states that research is needed to investigate whether these antimicrobials remain appropriatetreatment options or whether alternative interventions are more effective. But, the  presentation of MS is not good enough, due to this it is very hard  to understand MS.  I have some questions about MS.

1.  Author (s) should rewrite the content so the reader can understand the MS.2. It is very difficult to understand the Table-1, moreover some part of this table is missing in the MS.

Reviewer #2: This manuscript by MacCannN, Scott P, and colleagues entitled ''Antimicrobial agents for the treatment of enteric fever chronic carriage: A systematic review'' proposes to address the efficacy of different antimicrobials in treating enteric fever chronic carriage. In addition, this work shows the evidence for fluoroquinolones and amoxicillin as antimicrobial interventions for treatment of enteric fever carriage. The data collected in this systemic review could be of fundamental importance.

Some minor points to consider are included below:

In line 41: Please add comma(,) after 'identified'.

In line 218, please correct the data 'were' synthesized.

Table 1, table 2 and table 3 are incomplete. Please complete them.

6. PLOS authors have the option to publish the peer review history of their article (what does this mean?). If published, this will include your full peer review and any attached files.

Reviewer #1: No

Reviewer #2: No

---

## [Author Response · Author response to Decision Letter 0]

8 Jun 2022

Unfortunately we have been unable to successfully convert the manuscript from a doc to LaTeX format. Being unfamiliar with LaTeX it has taken us a significant number of hours to learn the basics of LaTex and convert the file from a word file into a .tex file in the correct format using the online instructions. Although able to correctly convert all the text (which we would be happy to upload and send) we have been unable to convert the tables and bibliography successfully despite a number of attempts. Currently, we are unable to progress further with the LaTeX manuscript conversion and therefore hope submitting in .doc file in the correct format is satisfactory. I have sent 2 emails to editors requesting advice regarding this but unfortunately have not heard back within the time period so am re-submitting the initial doc manuscript. 

I have provided a point-by-point response to the reviewers below: 

Reviewer #1: The MS reports on antimicrobial agents for the treatment of enteric fever chronic carriage: A systematic review. In this review the author presented about the efficacy of different antimicrobials in treating enteric fever chronic carriage. Author also states that research is needed to investigate whether these antimicrobials remain appropriate treatment options or whether alternative interventions are more effective. But, the presentation of MS is not good enough, due to this it is very hard to understand MS. I have some questions about MS.

1. Author (s) should rewrite the content so the reader can understand the MS.2. It is very difficult to understand the Table-1, moreover some part of this table is missing in the MS.

Apologies, there were some formatting errors in the initial manuscript submitted and therefore some of the information in the tables were not fully visible. This has been amended and all the data in all three tables are now fully visible. We hope this has significantly improved the presentation, and therefore understanding, of the manuscript in full. 

In addition, we requested a proof-read of the manuscript by an additional person, a clinical academic but not in the field of enteric fever research, to ensure the manuscript was understandable and presented well. They reviewed the manuscript and made a few small comments which have been incorporated in this revision, but commented that the overall presentation of the manuscript was good and easy to understand. 

If there are further areas of the manuscript that are still hard to understand we would be happy to amend these if advised where the issues are. 

Reviewer #2: This manuscript by MacCannN, Scott P, and colleagues entitled ''Antimicrobial agents for the treatment of enteric fever chronic carriage: A systematic review'' proposes to address the efficacy of different antimicrobials in treating enteric fever chronic carriage. In addition, this work shows the evidence for fluoroquinolones and amoxicillin as antimicrobial interventions for treatment of enteric fever carriage. The data collected in this systemic review could be of fundamental importance.

Some minor points to consider are included below:

In line 41: Please add comma(,) after 'identified'.

This has been added

In line 218, please correct the data 'were' synthesized.

This has been changed

Table 1, table 2 and table 3 are incomplete. Please complete them.

All three tables have been amended so all the data is fully visible. 

In addition please find our response to the Journal Requirements requested:

We have amended the manuscript as per the style requirements and template. 

Please note that in order to use the direct billing option the corresponding author must be affiliated with the chosen institute. Please either amend your manuscript to change the affiliation or corresponding author, or email us at plosone@plos.org with a request to remove this option.

University College London is listed as a chosen institute that has an Institutional Account with Plos. The corresponding author is affiliated with University College London and has a UCL email address. Please let us know if something additional is required. 

We note that you have stated that you will provide repository information for your data at acceptance. Should your manuscript be accepted for publication, we will hold it until you provide the relevant accession numbers or DOIs necessary to access your data. If you wish to make changes to your Data Availability statement, please describe these changes in your cover letter and we will update your Data Availability statement to reflect the information you provide.

All data is included within the manuscript itself therefore respository information for the data is not required. This has been amended in the submission.

---

## [Decision Letter · Decision Letter 1]

13 Jul 2022

Antimicrobial agents for the treatment of enteric fever chronic carriage: A systematic review

PONE-D-21-36297R1

Dear Dr. Naina McCann

We’re pleased to inform you that your manuscript has been judged scientifically suitable for publication and will be formally accepted for publication once it meets all outstanding technical requirements.

Kind regards,

Praveen Rishi, Ph.D., FAMI, FABMS

Academic Editor

PLOS ONE

Additional Editor Comments (optional):

Reviewers' comments:

Reviewer's Responses to Questions

**Comments to the Author**

1. If the authors have adequately addressed your comments raised in a previous round of review and you feel that this manuscript is now acceptable for publication, you may indicate that here to bypass the “Comments to the Author” section, enter your conflict of interest statement in the “Confidential to Editor” section, and submit your "Accept" recommendation.

Reviewer #1: (No Response)

Reviewer #2: (No Response)

2. Is the manuscript technically sound, and do the data support the conclusions?

Reviewer #1: Yes

Reviewer #2: (No Response)

3. Has the statistical analysis been performed appropriately and rigorously? 

Reviewer #1: No

Reviewer #2: (No Response)

4. Have the authors made all data underlying the findings in their manuscript fully available?

Reviewer #1: Yes

Reviewer #2: (No Response)

5. Is the manuscript presented in an intelligible fashion and written in standard English?

Reviewer #1: No

Reviewer #2: (No Response)

6. Review Comments to the Author

Reviewer #1: Author answered most of the questions and revised manuscript has been changed much of the content. I recommend it for acceptance.

Reviewer #2: (No Response)

7. PLOS authors have the option to publish the peer review history of their article (what does this mean?). If published, this will include your full peer review and any attached files.

Reviewer #1: No

Reviewer #2: No

---

## [Editor Report · Acceptance letter]

20 Jul 2022

PONE-D-21-36297R1 

Antimicrobial agents for the treatment of enteric fever chronic carriage: A systematic review 

Dear Dr. McCann:

I'm pleased to inform you that your manuscript has been deemed suitable for publication in PLOS ONE. Congratulations! Your manuscript is now with our production department. 

Kind regards, 

on behalf of

Prof. Praveen Rishi 

Academic Editor

PLOS ONE